# Exome-Wide Association Study Identified Clusters of Pleiotropic Genetic Associations with Alzheimer’s Disease and Thirteen Cardiovascular Traits

**DOI:** 10.3390/genes14101834

**Published:** 2023-09-22

**Authors:** Yury Loika, Elena Loiko, Irina Culminskaya, Alexander M. Kulminski

**Affiliations:** Biodemography of Aging Research Unit, Social Science Research Institute, Duke University, Durham, NC 27708, USA; elena.loiko@duke.edu (E.L.); irina.kulminskaya@duke.edu (I.C.); alexander.kulminski@duke.edu (A.M.K.)

**Keywords:** exome-wide association study, Alzheimer’s disease, pleiotropy, hierarchical cluster analysis, aging, cardio-metabolic traits, coronary heart disease, myocardial infarction, stroke, heart failure

## Abstract

Alzheimer’s disease (AD) and cardiovascular traits might share underlying causes. We sought to identify clusters of cardiovascular traits that share genetic factors with AD. We conducted a univariate exome-wide association study and pair-wise pleiotropic analysis focused on AD and 16 cardiovascular traits—6 diseases and 10 cardio-metabolic risk factors—for 188,260 UK biobank participants. Our analysis pinpointed nine genetic markers in the *APOE* gene region and four loci mapped to the *CDK11*, *OBP2B*, *TPM1*, and *SMARCA4* genes, which demonstrated associations with AD at *p* ≤ 5 × 10^−4^ and pleiotropic associations at *p* ≤ 5 × 10^−8^. Using hierarchical cluster analysis, we grouped the phenotypes from these pleiotropic associations into seven clusters. Lipids were divided into three clusters: low-density lipoprotein and total cholesterol, high-density lipoprotein cholesterol, and triglycerides. This split might differentiate the lipid-related mechanisms of AD. The clustering of body mass index (BMI) with weight but not height indicates that weight defines BMI-AD pleiotropy. The remaining two clusters included (i) coronary heart disease and myocardial infarction; and (ii) hypertension, diabetes mellitus (DM), systolic and diastolic blood pressure. We found that all AD protective alleles were associated with larger weight and higher DM risk. Three of the four (75%) clusters of traits, which were significantly correlated with AD, demonstrated antagonistic genetic heterogeneity, characterized by different directions of the genetic associations and trait correlations. Our findings suggest that shared genetic factors between AD and cardiovascular traits mostly affect them in an antagonistic manner.

## 1. Introduction

Alzheimer’s disease (AD) is an aging-related neurodegenerative disorder, with few interventions available to ameliorate its symptoms. As the number of AD-affected Americans is expected to increase rapidly, reaching about 12.7 million cases by 2050 [1], identifying the causes of Alzheimer’s disease would be beneficial for its prevention and treatment. Several hypotheses have been proposed to explain AD pathology, such as amyloid beta (Aβ) cascade, tau pathology, calcium, inflammatory, cholinergic, and oxidative stress hypotheses [2]. It is likely that many biological processes and pathways related to these hypotheses contribute to the initiation and progression of AD. In addition to the buildup of Aβ plaques and neurofibrillary tangles, the loss of neuronal synapse junctions, neuronal cell death, and brain atrophy can contribute to cognitive impairment underlying AD.

At this time, none of the aforementioned hypotheses alone have succeeded in explaining AD pathology. The amyloid beta hypothesis has been thoroughly investigated in the clinical treatment of AD, with a focus on targeting Aβ plaques. Notably, two drugs, lecanemab and aducanumab, have received approval from the FDA through the accelerated approval mechanism. Additionally, a third drug, donanemab, has shown promising results in clinical trials [3], but there is still no cure for AD at this time. This means that there are other processes accompanying the disease and contributing to its progression and severity. This includes the breakdown of the blood–brain barrier (BBB), the immune system response, reduced cerebral blood flow, altered adult neurogenesis and energy metabolism, cell membranes and the cell cycle, the cytoskeleton, and lipid and protein metabolism [4,5]. These processes can develop independently and/or be initiated by other age-related traits or events. The correlation of AD with these other traits may indicate common biological processes underlying them. Therefore, studying the patterns in the pleiotropic associations of AD and the other traits can facilitate identifying and understanding the biological mechanisms underlying AD pathogenesis.

Genome- and exome-wide association studies (GWAS and EWAS, respectively) are comprehensive tools designed to identify genes and related biological mechanisms associated with different traits. While conventional GWAS and EWAS focus on identifying genetic associations with single traits, pleiotropic GWAS and EWAS take a broader approach. They aim to simultaneously assess genetic associations with multiple traits and identify shared genetic components that contribute to the pleiotropy observed in epidemiological studies. Previous epidemiological studies that considered AD and other related traits helped identify several cardiovascular factors conferring AD risk. For instance, increased levels of LDL-C and TC are considered risk factors for AD [6,7], while increased levels of HDL-C were highlighted as protective [8]. Pleiotropic GWAS/EWAS provided an additional opportunity to identify several genetic components and respective biological mechanisms shared by those multiple traits [9,10].

Certain risk factors can form clusters, i.e., they increase or decrease AD risks simultaneously by acting in additive or synergistic manners. Several studies have already demonstrated the effects of such clusters [11,12]. For example, the clustering of vascular risk factors, such as midlife obesity, high SBP, and high total cholesterol levels, increased the risk of dementia and AD in an additive manner [12]. Clusters of hypertension and heart disease, diabetes, and current smoking also increased AD risk [11].

Many of the previous analyses of clustering of neurodegenerative conditions, including AD, used individual-level data for phenotypes as a distance measure [13,14,15,16,17]. In such types of cluster analyses, strongly correlated phenotypes usually cluster together. This approach is not quite efficient, however, in cases when the underlying biological mechanisms and related genetic components are not major contributors to the phenotypes’ correlations. In this study, we applied an approach in which the statistics of univariate and pleiotropic genetic associations were used for identifying clusters of AD and its risk factors. We believe that this approach can more accurately characterize complex roles of genetic and non-genetic factors in AD pathogenesis [18,19,20,21], including complex interactions, which can be attributed to different biological mechanisms driving antagonistic genetic heterogeneity [22,23]. Such heterogeneity was shown to be widespread in the genetics of lipid traits [24]. This phenomenon was also observed in pleiotropic associations with AD and some traits related to educational attainment and cardiovascular risk factors [25,26,27].

The goal of this study was to gain insights into pleiotropic predisposition to cardiovascular and AD risk factors and demonstrate complex clustering patterns of AD with 16 risk factors. We hypothesized that using summary statistics of pleiotropic associations among AD and related phenotypes (risk factors) could capture clusters of phenotypes based on the similarities among the genetic effects, rather than solely on the direct correlations among these phenotypes. This approach may facilitate the discovery of shared biological pathways. An additional goal was to investigate how antagonistic genetic heterogeneity shapes the clustering of AD-related phenotypes. For this purpose, we carried out a pair-wise pleiotropic exome-wide association study (~260 K common genetic variants) on predisposition to AD and on each of the 16 selected traits in a sample of ~190 K individuals from the UK Biobank. The analysis included six qualitative traits (coronary heart disease (CHD), diabetes mellitus (DM), stroke, myocardial infarction (MI), heart failure (HF), and hypertension (HT)), and ten other cardio-metabolic risk factors (blood glucose (BG), BMI, height, weight, four lipid traits (LDL-C, HDL-C, TG, and TC), and systolic (SBP) and diastolic (DBP) blood pressure). Fisher’s method was used to test the pleiotropic associations, and hierarchical cluster analysis was used to identify clusters of AD risk factors with similar genetic components.

## 2. Materials and Methods

### 2.1. Accession Numbers

This research has been conducted using data from the UK Biobank, a major biomedical database (http://www.ukbiobank.ac.uk/, accessed on 15 March 2021).

### 2.2. Study Cohorts

Data from the UK Biobank (UKB) [28,29] on individuals of Caucasian ancestry, men and women combined, were considered in the analyses. Table 1 provides the basic demographic information of the genotyped participants. After quality control of genetic and phenotypic information, 188,260 individuals remained in the subsequent analyses.

### 2.3. Genotypes

Exome sequencing data for ~200 K individuals were available from the UK Biobank Exome Sequencing Consortium [30]. In the analyses, we included directly genotyped markers (no imputation was performed) with minor allele frequency (MAF) greater than 0.5%, which resulted in 258,684 (~250 K) genetic variants. A total of 213,935 SNPs had a missing call rate better than 5%. For these SNPs, all individuals had a missing call rate better than 5%. Only these SNPs were considered for reporting the results of this study. We did not apply the Hardy–Weinberg equilibrium test at this stage because negligible deviation from Hardy–Weinberg equilibrium was still highly significant in the large UKB sample.

### 2.4. Phenotypes

We considered incident cases of AD and six other diseases, including DM, HT, CHD, MI, stroke, and HF (Table 1). All cases were defined by the UK Biobank based on the ICD-9 and ICD-10 codes. AD status was defined by using codes F00 and G30 according to the ICD-10 classification.

Ten quantitative phenotypes, BG (mg/dL), BMI (kg/m^2^), height (m), weight (kg), SBP (mmHg), DBP (mmHg), HDL-C (mg/dL), TG (mg/dL), LDL-C (mg/dL), and TC (mg/dL) were considered. Because longitudinal information (multiple measurements) was available for a small number of participants, we selected measurements based on the first examination for which information was available (Table 1) (see also Section 2.7). We found significant differences between the AD cases and controls in the qualitative phenotypes (assessed using the exact Fisher test) and the mean values for SBP (evaluated using the Wald test).

### 2.5. Correlations among Phenotypes

Information on the Pearson correlation coefficients between AD and each of the considered quantitative and qualitative phenotypes and their significance is given in Table 2. We observed a small positive significant correlation between AD and each of the six diseases. The largest and most significant correlation was between AD and HT (*r* = 0.029, *p* = 2.41 × 10^−23^). Among the quantitative traits, only weight, height, and SBP demonstrated small significant correlations with AD (Table 2).

Appendix A display the Pearson correlation coefficients among the various traits under consideration. Most correlations were statistically significant, with the exception of those involving stroke–height and the ones noted above for AD. Among the observed correlations, the highest were between LDL-C and TC (r = 0.95), CHD and MI (r = 0.58), SBP and DBP (r = 0.67), weight and BMI (r = 0.83), and weight and height (r = 0.54). Conversely, the most pronounced negative correlation was between HDL-C and TG (r = −0.44), as well as HDL-C and weight (r = −0.46).

### 2.6. Correlations among Summary Statistics

Table 2 also includes information on the Pearson correlation coefficients that were obtained by using summary statistics of the genetic associations with AD and each of the considered quantitative and qualitative phenotypes. Only the correlation coefficient between the summary statistics for AD and LDL-C was statistically non-significant. The correlation coefficients of the summary statistics for the other traits with AD were statistically significant. Moreover, for five traits (MI, stroke, HF, weight, and SBP), the correlation based on individual-level data and on summary statistics demonstrated opposite directions and were statistically significant. This demonstrates that correlations between the phenotypes at an individual level can be shaped by other factors in addition to genetic component(s).

### 2.7. Statistical Analyses

***The univariate unconditional analysis***—i.e., the univariate exome-wide association study (EWAS)—of the associations of each SNP with each quantitative phenotype or disease was performed using a linear or logistic regression model, as implemented in *plink 2.0* software. An additive genetic model was considered with the minor allele as an effect allele. All models in all analyses were adjusted for sex and age at the selected examination. Therefore, our EWAS provided 17 summary statistics (effect size, standard error, and *p*-value) for the estimates of the associations with seven diseases and ten continuous traits for each SNP.

***Pleiotropic meta-analysis*** was performed for the pairs of traits. We used an AD-centric approach when each pair included AD and one of the other traits. Given the small correlations among the traits (Table 2), Fisher’s method [31] was used for pleiotropic analysis. It combines *p*-values across phenotypes, disregarding the effect directions and correlations among them, and addresses the issue of multiple testing by increasing the number of degrees of freedom. Therefore, 16 pleiotropic meta-analyses were conducted.

### 2.8. Pleiotropic Associations

As Fisher’s method corrects pleiotropic *p*-values for testing multiple phenotypes, the traditional genome-wide (GW) level of significance, *p_GW_* = 5 × 10^−8^, was used to obtain conclusions about the significance of the pleiotropic effects. Here, we report GW significant pleiotropic SNPs when at least one of 16 pleiotropic tests attained GW significance, *p* < *p_GW_*, and the EWAS univariate associations with these traits demonstrated significant association at the levels of *p* < 5 × 10^−4^ for AD and *p* < 5 × 10^−2^ for the other traits. As this analysis was AD-centric, we applied a more stringent threshold for AD associations and a less stringent threshold for each of the other phenotypes.

### 2.9. Index SNPs and Gene Mapping

The NCBI dbSNP database and variant effect predictor from Ensembl (assembly GRCh38.p13) were used for mapping SNPs to genes. Outside of the *APOE* gene region, one index SNP was selected per locus. Multiple genes were reported when several genes overlapped. The index SNPs were selected based on the most significant association (smallest *p*-values) within each genetic locus. Furthermore, the selection of index SNPs took into consideration the proximity of the MAF from the UKB exome chip to the MAF found in the UK10 K TWINS and/or 1000 G reference datasets, as the MAF for some SNPs substantially differed. SNPs that met the Hardy–Weinberg threshold *p*-value = 1 × 10^−40^, were retained. We used a small *p*-value for the Hardy–Weinberg cut-off because even very small deviation from Hardy–Weinberg equilibrium was highly significant in the large UKB sample.

### 2.10. Cluster Analysis

Hierarchical cluster analysis can be applied if a distance measure is defined. Previous studies [11,12,13,14,15,16,17] used individual-level data of phenotype measurements as a distance measure, which assesses the similarity of phenotypes, including their genetic effects, effects of exogenous exposures, and their interaction. In this study, we proposed using genetic associations with phenotypes as a distance measure. This approach is more appropriate for gaining insight into the similarities related to genetic components rather than exogenous exposures, considering that the contributions of genetic effects and exogenous exposures on the correlations among the phenotypes can be substantially different (see Table 2).

In this study, SNP–phenotype clusters were identified, leveraging the results of our univariate and AD-centric pair-wise pleiotropic analyses. First, we identified patterns characterized by the relative directions of the associations of the same alleles with AD and each of the other phenotypes and the significance of the pleiotropic associations. The relative directions were defined by the sign of a product of the effect sizes of the associations of SNPs with AD and the other phenotype in a pair. For instance, if the associations of SNPs with these phenotypes were of the same (opposite) directions, such SNPs had the same (different) pattern.

These patterns can be subdivided further into clusters based on the *p*-values of the associations of SNPs with phenotypes. To improve the resolution of the cluster analysis, we used *p*-values from the univariate analyses of non-AD phenotypes and selected pleiotropic SNPs for which the significance of their associations with AD was *p* < 5 × 10^−4^. We further categorized minus-log-transformed *p*-values from the univariate associations of non-AD phenotypes, −*log(p_uni_)*, into five categories, denoted as 0 for *p* > 0.05, 1 for 5 × 10^−4^ ≤ *p* < 5 × 10^−2^, 2 for 5 × 10^−6^ ≤ *p* < 5 × 10^−4^, 3 for 5 × 10^−8^ ≤ *p* < 5 × 10^−6^, and 4 for *p* < 5 × 10^−8^. This procedure created an *A_ij_* matrix for the *i*th SNP and *j*th phenotype, in which the sign of each element was determined by the relative directions of the SNP associations with AD and a respective trait, and its magnitude was defined by the aforementioned categories.

Two approaches were used for cluster analysis. Both approaches utilized hierarchical cluster analysis with Euclidian measure and Ward’s method. Under this method, associations are clustered by minimizing the distance measure inside the clusters and maximizing this measure between clusters. Our main approach was applied to matrix *A_ij_* as implemented by the *hclust* function in R. This approach estimates clusters based on the similarity of significances and sign-directions of the SNP–phenotype associations. The ad-hoc height cut-off threshold for selecting the clusters was equal to five. This analysis was validated using an approach based on Pearson’s correlation as a distance measure. It was implemented using the *pheatmap* function in R package *ecodist*. This approach estimates clusters based on the collinearity of phenotypes as vectors of the associations with selected SNPs (collinearity of columns of matrix *A_ij_*) and on the collinearity of the associations of selected SNPs with the considered phenotypes (collinearity of rows of matrix *A_ij_*).

## 3. Results

Figure 1 represents a flowchart of the analyses performed in this study.

### 3.1. Univariate Associations from EWAS

A univariate EWAS was performed for seven diseases (DM, HT, CHD, MI, HF, stroke, and AD) and ten quantitative phenotypes (BG, BMI, weight, height, SBP, DBP, HDL-C, TG, LDL-C, and TC) in a sample of 188,260 UK biobank participants of European ancestry, for men and women combined (see Section 2).

For the diseases, the only GW significant associations were observed with AD in the *APOE* gene cluster on chromosome 19 for rs112849259 and rs741780 *TOMM40* SNPs and rs440446 and rs429358 *APOE* SNPs (Table 3 and Appendix A). Also, for SNPs in the *APOE* gene cluster, we identified GW significant associations with at least one of the lipid traits: HDL-C (three SNPs), LDL-C (six SNPs), TG (five SNPs), and TC (six SNPs). In total, this cluster of genes harbored 24 GW significant univariate SNP–trait associations (Appendix A).

Outside of the *APOE* gene cluster, three SNPs demonstrated five GW significant associations with BMI (rs28688376), LDL-C, and TC (rs11244035, rs28997580) (Table 4 and Appendix A).

In total, 29 GW significant univariate associations were identified.

### 3.2. Pleiotropic AD-Centric Pair-Wise Associations

We identified 51 GW significant pleiotropic associations (see Section 2) (Appendix A). They included 20 associations in the *APOE* gene cluster, which were not driven by the GW significance of the associations with AD. Four SNPs in the *APOE* and *TOMM40* genes, which were associated with AD at the GW level, showed 19 GW significant univariate associations with non-AD phenotypes, totaling 19 GW significant pleiotropic associations. These SNPs were associated with BMI, weight, and DBP (rs112849259 and rs429358); height and TC (rs741780 and rs440446); HDL-C (rs112849259 and rs741780); LDL-C (rs741780); DM (rs112849259, rs440446, and rs429358); HT (rs440446); and CHD and MI (rs429358). Also, the remaining SNPs from the *APOE* gene cluster showed five additional significant pleiotropic associations with AD and BMI, weight, CHD, and MI (rs7412); HDL-C (rs28399637), while these SNPs did not attain GW significance in the univariate analysis. In total, 44 [=20 + 19 + 5] GW significant pleiotropic associations were found for the SNPs mapped to the *APOE* gene cluster. Outside of the *APOE* cluster, we identified seven GW significant pair-wise pleiotropic associations. These included five SNP–trait pairs with GW significant univariate effects (Table 4 and Appendix A) and two additional pleiotropic associations, including BMI (rs28688376) and HDL-C (rs4775613).

We also identified 11 pleiotropic associations at the suggestive-effect level 5 × 10^−8^ < *p* < 5 × 10^−6^. They included associations of rs4775613 with SBP and TC; rs80168591 with HDL-C and stroke; rs28399637 with BMI, weight, and CHD; rs7412 with SBP and HT; as well as rs5167 with height and TC. Thus, in total, we found 62 (=51 + 11) pair-wise pleiotropic associations for 13 SNPs and 14 traits at *p* < 5 × 10^−6^ (Appendix A). None of the pleiotropic associations attained even a suggestive-effect level for BG and HF, and only one for stroke.

### 3.3. Clusters of Pleiotropic Associations

For cluster analysis (see Section 2.10), we used 61 pleiotropic associations, excluding unique association with stroke (Appendix A). Matrix *A_ij_* for this analysis included information on 13 SNPs and 13 phenotypes (Appendix A). Hierarchical cluster analysis using the Euclidian measure and Ward’s method provided two-dimensional dendrogram in the SNP–phenotype domain (Figure 2). Phenotype-specific one-dimensional dendrogram exemplified seven clusters for 13 phenotypes, i.e., (1) (LDL-C, TC), (2) TG, (3) (CHD, MI), (4) height, (5) (HT, SBP, DBP, DM), (6) HDL-C, and (7) (weight, BMI) (Figure 2). Pearson’s correlation approach (Appendix A) provided similar results, except that a pair of (HT, SBP) phenotypes was clustered with (CHD, MI) rather than with (DBP, DM). This discrepancy is likely due to diminished power to identify clusters using different distance measures, given the moderate *p*-values from the univariate associations with these six phenotypes (Appendix A). Both approaches of cluster analysis aggregated SNPs into the same eight clusters (Figure 2 and Appendix A).

Figure 2 shows that the largest group of GW significant univariate (8 for LDL-C and 8 for TC) and pleiotropic (9 for LDL-C and 10 for TC) associations was observed for the (LDL-C, TC) cluster. The second largest group was observed for TG (five univariate and five pleiotropic GW significant associations), followed by HDL-C (three univariate and seven pleiotropic GW significant associations) and BMI (one univariate and four pleiotropic GW significant associations) + weight (four pleiotropic GW significant associations) clusters. Two distinct clusters emerged from the analysis, namely (LDL-C, TC) and (BMI, weight), which were composed of traits with the largest Pearson correlation coefficients of r = 0.952 and r = 0.833, respectively, evaluated using individual-level data (Appendix A). This arrangement follows the anticipated clustering pattern where correlated traits are grouped together. However, in contrast, height and weight, despite having a substantial significant correlation of r = 0.538, were placed in separate phenotypic clusters.

### 3.4. Antagonistic Genetic Heterogeneity

Figure 2 emphasizes the complex forms of genetic heterogeneity when the same alleles from different SNPs can have the same (e.g., ε2-encoding rs7412 SNP) or opposite (e.g., minor allele of rs28997580) directions of the associations with AD and other phenotypes in a pair, e.g., (AD, LDL-C). This pattern, i.e., two sets of SNPs, was observed for each of the eight phenotypes from six phenotypic clusters: LDL-C + TC, TG, HDL-C, height (H), HT + SBP, and CHD.

Figure 3 provides further insight into the phenomenon of antagonistic genetic heterogeneity. It includes the results of the significant pleiotropic associations for the traits that demonstrated significant correlations with AD. Associations with lipid traits, BMI, and DBP were excluded because of their nonsignificant correlations with AD (Table 2). Each cell in Figure 3 displays the sign of a product of effect sizes for SNP–AD and SNP–trait associations and the Pearson correlation coefficient (calculated by using individual-level data) of AD and the corresponding trait. Red (blue) cells, which correspond to the negative (positive) sign of the product mentioned just above, represent pleiotropic associations with antagonistic (non-antagonistic) genetic effects. From this figure, it follows that 33% (7 out of 21) of the pleiotropic associations in 3 of 4 phenotypic clusters (75%) demonstrated antagonistic genetic heterogeneity, i.e., alleles, which demonstrated detrimental association with AD, also demonstrated the opposite sign direction of their association with the considered trait compared to the correlation of that trait with AD. Two traits demonstrated a more homogeneous pattern. Significant associations with DM demonstrated antagonistic pleiotropic effects only, while all significant pleiotropic associations, which included weight, were non-antagonistic ones (Figure 3).

## 4. Discussion

In this EWAS, we performed AD-centric pair-wise pleiotropic and cluster analyses of the associations of SNPs with AD and 16 phenotypes of cardiovascular and AD risk factors.

### 4.1. AD-Centric Pair-Wise Pleiotropic Associations

AD-centric pleiotropic analysis considered the associations of SNPs with AD and one of the 16 phenotypes. This analysis identified 13 SNPs with genome-wide significant pleiotropic associations (Figure 2). Nine SNPs were mapped to the *APOE* gene cluster on chromosome 19 (including *APOE*, *TOMM40*, *APOC2*/*APOC4*, *PVRL2* (*NECTIN2*), *BCAM*, and *CBCL* genes), which is a well-known genetic risk factor for AD. Four other SNPs were from *CDK11* (rs28688376), *OBP2B* (rs11244035), *TPM1* (rs4775613), and *SMARCA4* (rs28997580) gene loci. These SNPs were associated with AD at *p* < 5 × 10^−4^ and attained genome-wide significance (*p* < 5 × 10^−8^) in the pleiotropic analysis.

The ***CDK11B*** (cyclin-dependent kinase 11B) gene plays a role in ***cell apoptosis***. Recent studies have demonstrated that *CDK11* showed an altered expression in AD vulnerable neurons, which may be related to APP signaling processes [34,35]. A predominantly increased expression of *CDK11* was observed in the cytoplasm of neuronal cells in AD cases, while it was expressed specifically in the nuclei of post-mitotic neurons in most controls [34]. CDK11 is regulated by checkpoint kinase 2 (CHK2), which phosphorylates tau at an AD-related site, enhancing tau toxicity [36,37], which demonstrates an additional link of the *CDK11B* gene to AD. Altered expression of *CDK11B* can be a cause of neuronal apoptosis and, consequently, of decreased brain weight through this process. This suggestion is consistent with our results that the minor allele of rs28688376 from the *CDK11B* gene was associated with an increased risk of AD and decreased levels of weight and BMI.

***TPM1****, TPM3,* and *TPM4* genes encode tropomyosin isoforms in neuronal cells. *TPM1* products were found in the presynaptic compartment of the central nervous system (CNS) neurons [38]. Tropomyosins (Tpm) actin-binding proteins ***stabilize the actin filaments,*** which play a key role in the synaptic function of the CNS and mediate processes of memory and learning [39]. Microtubules, neurofilaments, and microfilaments (actin filaments) form the cytoskeleton of neurons. ***Dysregulation*** of tropomyosin and the actin ***cytoskeleton can induce synapse loss***, which takes place in the early stages of AD pathology. Tropomyosin participation in the neurofibrillary pathology of AD was immunochemically demonstrated [40]. A significant increase in tropomyosin-1 abundance in the platelets of AD female patients was recently observed [41], in addition to a proteomics study that showed *TPM1* gene products increasing in the white matter of AD patients when compared to controls [42]. Recently, it has been demonstrated that *TPM1* plays a key role in cardio-genesis and cardiovascular disorders [43], providing a link to the associations of SNPs from this gene with SBP observed in this study.

The protein encoded by the ***SMARCA4*** gene is a member of the SWI/SNF family of proteins, which regulate gene activity by a chromatin remodeling and are involved in repairing damaged DNA, replicating DNA, and controlling the growth, division, and differentiation of cells. *SMARCA4* is in the same gene region as the *LDLR* gene. ***LDLR*** is a key regulator of cholesterol metabolism and encodes the protein involved in receptor-mediated endocytosis of low-density lipoprotein cholesterol [44]. The genome-wide-significant association of rs2569540 from the *LDLR* gene with AD was demonstrated in a recent GWAS study [45].

The ***OBP2B*** gene plays a role in chemosensory behavior and the perception of smell. Impaired sense of smell, or olfactory dysfunction, is often seen as an early indicator of AD pathology in the brain. A recent study indicated that individuals with olfactory dysfunction also exhibit changes in the blood levels of LDL-C and TC [46]. The current study aligns with these findings and adds additional insight, suggesting how blood cholesterol levels and AD-related changes could be intertwined. Specifically, our results suggest that olfactory dysfunction, which is considered as an early manifestation of AD, can be related to altered levels of LDL-C and TC in the blood. This link highlights the need for further investigation to determine whether AD might play a causal role in altering lipid metabolism.

Our findings suggest the involvement of other mechanisms contributing to the disease pathogenesis in addition to the amyloid-beta mechanism. Specifically, they highlight actin filaments, chromatin remodeling, neuronal apoptosis, and lipid abnormalities as potential targets for the development of new AD drugs.

### 4.2. Clustering of Genome-Wide Significant Pair-Wise AD-Centric Pleiotropic Associations

In previous studies [11,12,13,14,15,16,17], the clustering of neurodegenerative conditions, including AD, was considered based on correlations among phenotypes. Our cluster analysis added a pleiotropic genetic component and, therefore, it can help in gaining further insights into the biological mechanisms associated with the pleiotropic effects. Such biological mechanisms may or may not be those which drive correlations among phenotypes. In this study, phenotypes belong to the same cluster if the associations of the SNPs with these phenotypes and AD demonstrated similarity defined by the same effect directions and the level of statistical significance (see Section 2.10). Thus, clustering is considered in a two-dimensional SNP/gene and phenotype domain. Similarity within phenotypic clusters suggests underlying biological mechanisms, which may contribute to AD. Our cluster analysis identified eight clusters of SNPs and seven clusters of phenotypes, which demonstrated similar SNP–phenotype associations (Figure 2). Cluster analysis was harmonized by considering alleles predisposed to AD as effect alleles.

We found that lipids defined three clusters: one cluster was defined by LDL-C and TC, while TG and HDL-C were in two separate clusters. This finding suggests at least partial independence of the biological mechanisms underlying the pleiotropic associations of each of these lipid traits and AD. This result corroborates previous studies [25,47], which reported partially independent mediation of the genetic associations with AD through LDL-C, TG, and HDL-C for SNPs inside and outside of the *APOE* gene region.

A CHD and MI cluster was defined by significant associations of SNPs from the *APOE* gene locus (Figure 2 and Appendix A). Interestingly, while most alleles with elevated risks for CHD and MI (rs7412_C: *β*_CHD_ = 0.12, *p* = 8.25 × 10^−8^; and rs423958_c: *β*_CHD_ = 0.06, *p* = 2.90 × 10^−4^) were associated with higher risk of AD (rs7412_C: *β*_AD_ = 0.86, *p* = 1.87 × 10^−5^; and rs423958_c: *β*_AD_ = 1.37, *p* = 4.63 × 10^−71^), rs283813 (*PVRL2* gene) demonstrated the opposite relationship, i.e., rs283813_a was favorably associated with CHD (*β*_CHD_ = −0.05, *p* = 3.01 × 10^−2^) but adversely with AD (*β*_AD_ = 0.49, *p* = 7.16 × 10^−5^). This finding demonstrates that alleles associated with increased risks of CHD can be associated with either a higher or lower risk of AD. This is in line with antagonistic pleiotropic associations identified for AD and other cardiovascular factors, such as DM, that likely reflects contributions of different biological processes [27,48,49].

Four traits (HT, SBP, DBP, DM) defined a cluster with a complex pattern of associations. As they are risk factors for CHD and MI, they are typically considered together with these diseases. Meanwhile, in this study, these phenotypes belong to two different clusters, as there is more similarity within the clusters than among the clusters. All six phenotypes and height would belong to the same cluster if the cut-off of the cluster height level was increased to 7 (Figure 2, see the vertical axis with the numbers in the top left corner). Interestingly, SNPs encoding *APOE* ε2 and ε4 alleles were associated with two different subclusters, (HT, SBP) and (DBP, DM), respectively. This finding emphasizes that the pleiotropic associations of these *APOE* alleles have different etiologies.

Height and (BMI, weight) defined two non-overlapping clusters, indicating a role of independent mechanisms contributing to height and BMI or weight and AD.

Three phenotypes (BG, HF, and stroke) did not define any phenotypic cluster because of the lack of significant genetic associations that could be used for clustering.

### 4.3. Antagonistic Genetic Heterogeneity Was Observed for AD-Centric Pleiotropic Associations with Five Traits

Antagonistic genetic heterogeneity is characterized by misalignment of the signs of the product of the effects of genetic associations with different phenotypes and the correlations among these phenotypes [22,23,24,25,26]. This heterogeneity in our study was seen in the associations of seven SNPs with five traits in three phenotypic clusters (Figure 3). For each cluster of phenotypes (apart BMI, weight and DBP, DM), there were two sets of SNPs, which were characterized by antagonistic and non-antagonistic genetic effects in the associations with AD and the second phenotype in a pair (Figure 3). For example, rs283813 showed antagonistic genetic heterogeneity because it was adversely associated with AD and favorably with CHD, despite a significant positive correlation between AD and CHD (Table 2). The rs429358 did not show antagonistic heterogeneity with AD and weight because opposite-direction associations of this SNP with these phenotypes were aligned with a negative correlation between AD and weight.

Antagonistic genetic heterogeneity can also be viewed as a source of SNP/gene-phenotype clustering. For example, antagonistic genetic heterogeneity differentiates a cluster of height from that of BMI and weight because the height cluster includes antagonistic heterogeneity whereas the latter does not.

Antagonistic genetic heterogeneity plays a substantial role in the genetic associations with AD and its risk factors. This heterogeneity indicates that an effect allele from an SNP can confer the risk of one trait while being favorably associated with the other traits. Such a relationship likely indicates the role of different biological processes associated with the genes and pathways mapped to such SNPs on AD and its risk factors. For example, for DM, all significant pleiotropic associations were antagonistic. The alleles of SNPs with a significant adverse effect on DM were favorably associated with AD. This finding corroborates previous results [27,48,49] and highlights the complex contribution of genetic components to these diseases.

### 4.4. Protective Effect of Higher BMI Level against AD Is Related to High Weight

Previous studies demonstrated the protective effect of a high BMI level in late life against AD [20,21]. BMI is defined by two phenotypes, weight and height. Our cluster analysis showed that weight and height belong to different clusters. Moreover, SNPs that demonstrated genome-wide significant pleiotropic associations with BMI and weight were not significantly associated with height and vice versa. This finding suggests that protective effects of a higher BMI level against AD are related to higher weight rather than smaller height. While an increased BMI resulting from weight gain is associated with a higher risk of cardiovascular diseases (such as CHD, MI, and hypertension) and diabetes in midlife, a decreased BMI resulting from weight loss in later life can increase the risk of AD. This risk is attributed to the deteriorated functions of an aging organism, which include impaired metabolism, reduced cholesterol levels, decreased muscle mass, frailty, and sarcopenia, among other factors.

## 5. Conclusions

This study reports four novel genetic loci showing pleiotropic associations with AD and at least with one of 13 cardiovascular and AD risk factors. Genes harboring these loci are involved in cell apoptosis (*CDK11*), the stabilization of actin filaments, i.e., the cytoskeleton of neurons, (*TPM1*), the regulation of chromatin remodeling (*SMARCA4*), and chemosensory behavior and sensory perception of smell (*OBP2B*). No pleiotropic associations were identified for BG, HF, or stroke. Leveraging information on univariate and pleiotropic genetic associations, we found seven clusters in the domain of 13 SNPs and 13 phenotypes. Nine of thirteen pleiotropic SNPs were mapped to the *APOE* gene cluster (Figure 2).

AD-centric pleiotropic analysis confirmed partially independent mechanisms of pleiotropic associations with AD and lipid traits; although LDL-C and TC defined one phenotypic cluster, TG and HDL-C formed independent clusters because of the different patterns of the genetic associations.

Our analysis identified that 61% (9 of 13) of the considered traits demonstrated antagonistic pleiotropic AD-centric associations. Our cluster analysis found that weight, which was clustered with BMI, but not with height, defined pleiotropy of BMI and AD. SNPs identified by significant pleiotropic associations with AD and DM were from the *APOE* gene cluster, which corroborates previous findings.

Our cluster analysis in a two-dimensional SNP/gene-phenotype domain highlighted a more complex role of genetic and non-genetic factors in AD pathogenesis than the cluster analyses based on phenotype correlations did. Indeed, assuming that the clustering of phenotypes is related to their correlations, it is expected that correlated phenotypes should be in the same cluster. For LDL-C and TC (Pearson’s correlation coefficient, r = 0.952), BMI and weight (r = 0.833), and CHD and MI (r = 0.581), this was the case, as each pair of these phenotypes was in its own cluster. This was not the case, however, for the other correlated phenotypes, such as, for example, height and weight (r = 0.538), HDL-C and weight (r = −0.462), and HDL-C and TG (r = −0.439) (see Appendix A) because these phenotypes were in different clusters. Accordingly, our cluster analysis approach was capable of capturing the biological mechanisms relevant to pleiotropy underlying AD and other age-related phenotypes that may be not readily available from analyses relying on information about the correlations of phenotypes. Defining more homogeneous patterns of phenotypes and genetic factors can aid in the identification of more homogenous groups of individuals who are at varying risks of AD. Subsequent research can facilitate the identification of genetic and non-genetic components, contributing to the pleiotropic effects associated with AD. Such insights can be valuable for developing comprehensive treatment approaches that combine drug therapy and lifestyle interventions.

The proposed approach can be applied to published summary statistics of genetic associations with different traits. Such an extension can help identify phenotypes that share genetic components and, therefore, overlapping biological mechanisms between different complex traits.

This study has some limitations. First, the analyses were performed for the entire sample but not separately for groups that are at different risk levels of AD, such as, for instance, within each sex group and/or at different ages. Second, we used a threshold *p*-value = 5 × 10^−4^ for associations with AD, which substantially limited the number of significant pleiotropic associations/SNPs. Third, a small number of SNPs (because of the second limitation) were available for hierarchical cluster analysis, which limited the precision of identifying phenotype clustering.

## Figures and Tables

**Figure 1 genes-14-01834-f001:**
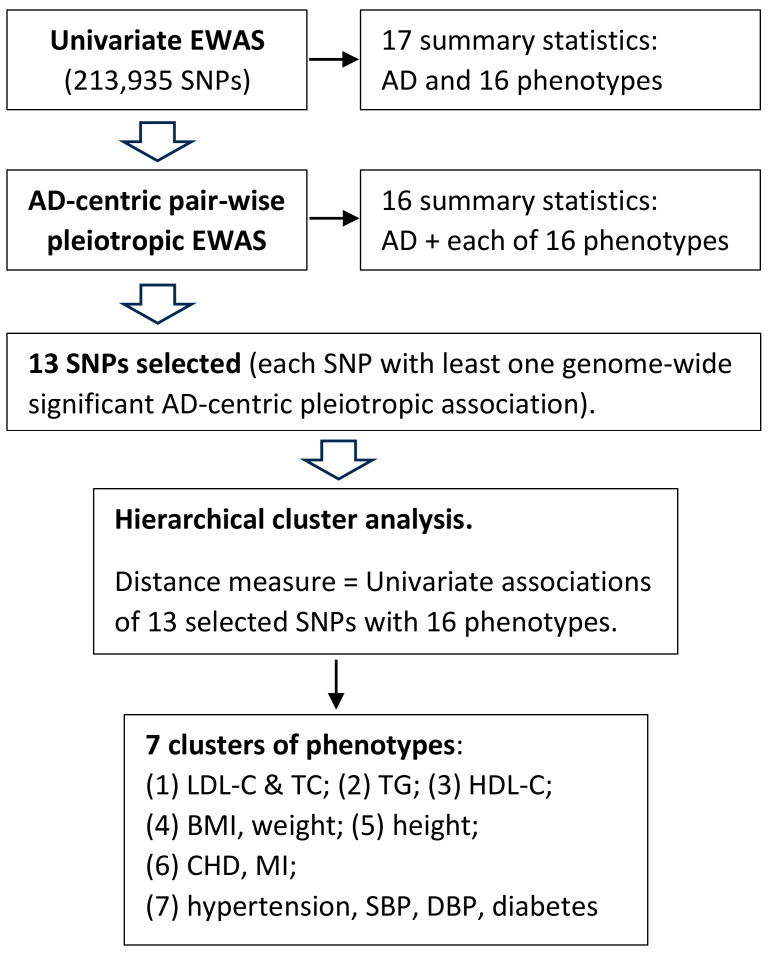
Flowchart of the analyses of this exome-wide association study (EWAS).

**Figure 2 genes-14-01834-f002:**
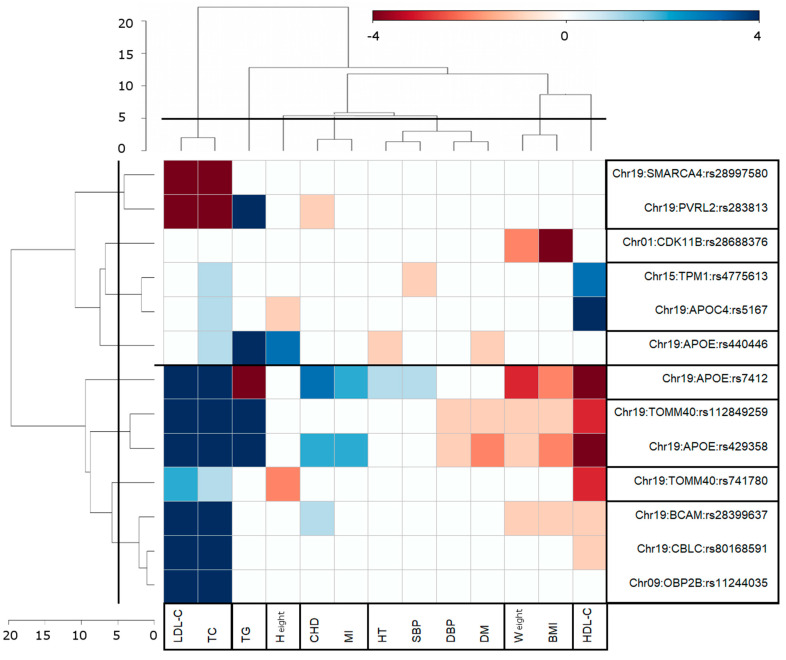
Eight clusters of SNPs and seven clusters of phenotypes based on significant pleiotropic pair-wise associations with AD and each of the other 13 age-related traits. Blue (red) color and its shades shows SNP–phenotype associations with the same (opposite) effect direction compared to the respective SNP–AD associations. Alleles, which demonstrated detrimental association with AD (positive betas), were considered effect alleles. Boxes show SNP and phenotypic clusters of pleiotropic associations, which were selected at a score (height) level of 5, as represented by axes with numbers on the horizontal (phenotypes) and vertical (SNPs) dendrograms. The *x*-axis shows the phenotypes, while *y*-axis refers to the SNPs and their respective genes. The horizontal solid line in the center shows the separation of the SNP clusters into two groups based on their significance and the effect directions of their associations with the first phenotypic cluster (LDL-C, TC). Hierarchical cluster algorithm with Euclidian measure and Ward’s method as implemented in R function hclust was used for this plot.

**Figure 3 genes-14-01834-f003:**
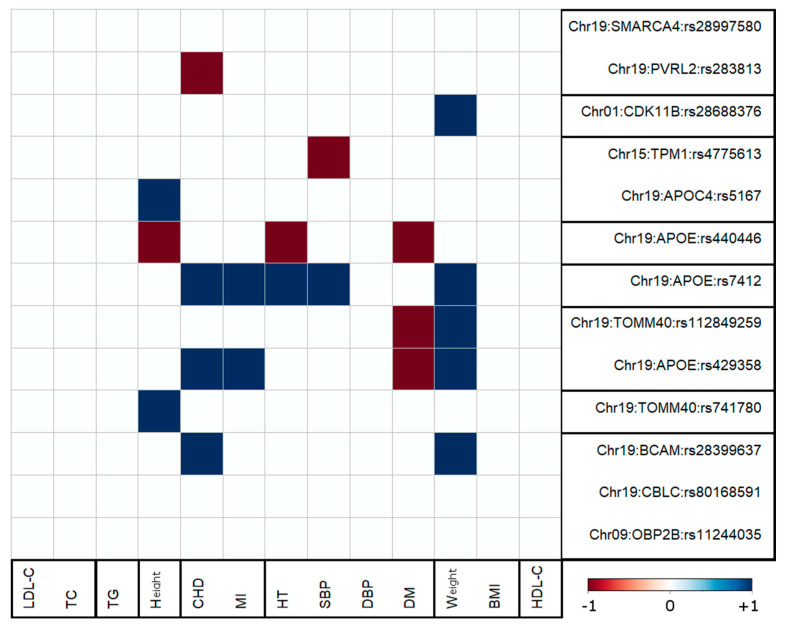
Antagonistic genetic heterogeneity of significant pleiotropic pair-wise associations with AD and each of the other 13 age-related traits. Blue (red) color shows non-antagonistic (antagonistic) genetic heterogeneity. The results are presented for phenotypes that exhibit significant correlations with AD. Therefore, associations involving lipids, BMI, and DBP were excluded from this figure (see Table 2). Refer to Figure 2 caption for other notations.

**Table 1 genes-14-01834-t001:** Basic demographic information of the UK biobank participants with whole exome genotyping information.

AD Status	PAR	N		Women	BC Range		LS	Age		Mortality
	VALUE (SD)	188,260		103,736 (55.10)	1936–1970		65.21 (8.0)	56.70 (8.03)		899 (0.48)
	**Quantitative traits**									
	AD	DM		HT	CHD		MI	STROKE		HF
Cases	361	53 (15%)		189 (52%)	89 (25%)		37 (10%)	30 (8%)		20 (6%)
Controls	187,899	11,015 (6%)		45,349 (24%)	16,327 (9%)		5834 (3%)	4589 (2%)		3236 (2%)
*p*-value		1.33 × 10^−9^		<2.2 × 10^−16^	<2.2 × 10^−16^		3.90 × 10^−10^	9.97 × 10^−9^		6.867 × 10^−6^
	**Quantitative traits**									
	BG (mg/dL)	BMI (kg/m^2^)	Height (cm)	Weight (kg)	SBP (mmHg)	DBP (mmHg)	HDL-C (mg/dL)	LDL-C (mg/dL)	TC (mg/dL)	TG (mg/dL)
Cases	93.36 (19.93)	27.01 (4.90)	167.18 (9.17)	75.62 (15.40)	146.18 (19.58)	81.79 (10.96)	57.77 (16.30)	133.77 (35.72)	216.97 (47.43)	149.26 (78.77)
Controls	92.13 (21.19)	27.34 (4.73)	168.67 (9.24)	78.03 (15.83)	139.78 (19.58)	82.13 (10.65)	56.49 (14.82)	138.07 (33.44)	221.19 (43.99)	154.26 (89.82)
*p*-value	5.89 × 10^−1^	5.05 × 10^−1^	1.61 × 10^−1^	1.72 × 10^−1^	3.46 × 10^−2^	9.04 × 10^−1^	4.77 × 10^−1^	2.68 × 10^−1^	3.96 × 10^−1^	5.54 × 10^−1^

PAR = the name of characteristics/parameter; SD = standard deviation. N = sample size; Women = number of women (%); BC Range = range of birth years; LS = life span in years; age = mean age measured in years at the selected examination of quantitative traits. *p*-values for qualitative and quantitative phenotypes were assessed by using the exact Fisher and Wald test, respectively. AD = Alzheimer’s disease; DM = diabetes mellitus; HT = hypertension; CHD = coronary heart disease; MI = myocardial infarction; STROKE = stroke; HF = heart failure. BG = blood glucose; BMI = body mass index; SBP = systolic blood pressure; DBP = diastolic blood pressure; HDL-C = high-density lipoprotein cholesterol; LDL-C = low-density lipoprotein cholesterol; TC = total cholesterol; TG = triglycerides.

**Table 2 genes-14-01834-t002:** Pearson correlation coefficient between Alzheimer’s disease and each of the considered phenotypes in the UK Biobank based on individual-level data and summary statistics of genetic associations.

	Individual-Level Data		Summary Statistics	
Phenotype	r	P	r	P
HT *	0.0290	2.41 × 10^−23^	0.0352	<2.2 × 10^−16^
CHD *	0.0248	1.82 × 10^−17^	0.0145	2.15 × 10^−11^
MI *	0.0180	6.52 × 10^−10^	−0.0169	4.67 × 10^−15^
STROKE *	0.0166	1.19 × 10^−8^	−0.0174	8.10 × 10^−16^
DM *	0.0164	1.76 × 10^−8^	0.0365	<2.2 × 10^−16^
HF *	0.0128	1.08 × 10^−5^	−0.0358	<2.2 × 10^−16^
BG	0.0025	3.82 × 10^−1^	−0.0069	1.55 × 10^−3^
BMI	−0.0031	2.82 × 10^−1^	0.0514	<2.2 × 10^−16^
Height *	−0.0070	1.61 × 10^−2^	−0.0700	<2.2 × 10^−16^
Weight *	−0.0066	2.30 × 10^−2^	0.0092	2.03 × 10^−5^
SBP *	0.0143	9.34 × 10^−7^	−0.0082	1.41 × 10^−4^
DBP	−0.0014	6.35 × 10^−1^	0.0164	2.87 × 10^−14^
HDL-C	0.0038	1.96 × 10^−1^	−0.0074	5.84 × 10^−4^
LDL-C	−0.0057	5.17 × 10^−2^	−0.0029	1.87 × 10^−1^
TC	−0.0042	1.48 × 10^−1^	−0.0086	7.62 × 10^−5^
TG	−0.0024	4.00 × 10^−1^	−0.0078	3.41 × 10^−4^

r = Pearson correlation coefficient; P = *p*-values; * = significant correlation based on individual-level data. DM = diabetes mellitus; HT = hypertension; CHD = coronary heart disease; MI = myocardial infarction; STROKE = stroke; HF = heart failure. BG = blood glucose; BMI = body mass index; SBP = systolic blood pressure; DBP = diastolic blood pressure; HDL-C = high-density lipoprotein cholesterol; LDL-C = low-density lipoprotein cholesterol; TC = total cholesterol; TG = triglycerides.

**Table 3 genes-14-01834-t003:** Associations with risk of Alzheimer’s disease: SNPs from 4 genetic loci and *APOE* gene region.

N	Gene(s) ^1^	SNP ^2^	Chr	Location, Base Pairs GRCh38	Ref /Alt	MAF obs (%)	P_HWE_	Function	Beta	SE	P	Gene _GRASP_	SNP _GRASP_	r^2^	P_GRASP_	PMID
1	CDK11B (MMP23B)	rs28688376	1	1,637,577	T/c	26.1	6 × 10^−36^	intron	0.298	0.079	1.56 × 10^−4^					
2	OBP2B	rs11244035	9	133,205,932	C/t	10.2	2 × 10^−30^	missense	0.393	0.109	3.23 × 10^−4^	ABO	rs8176694	0.02	1.3 × 10^−2^	20061627
3	TPM1	rs4775613	15	63,056,897	A/g	42.8	3 × 10^−2^	5′UTR	−0.334	0.078	1.99 × 10^−5^	RAB8B	rs10519190	0.00	2.2 × 10^−4^	17998437
4	SMARCA4	rs28997580	19	11,013,062	C/t	0.9	3 × 10^−2^	synonymous	0.960	0.255	1.64 × 10^−4^	LDLR	rs2569540	0.00	1 × 10^−9^	35589863
	APOE		19													
5	CBLC	rs80168591	19	44,781,370	G/a	1.4	0.87	splice	0.911	0.205	8.54 × 10^−6^	CBLC	rs899087	0.00	5.1 × 10^−5^	22832961
6	BCAM	rs28399637	19	44,820,881	G/a	31.5	5 × 10^−8^	intron	0.382	0.076	4.71 × 10^−7^	BCAM	rs2927480	0.21	5.0 × 10^−49^	21460841
7	PVRL2	rs283813	19	44,885,917	T/a	6.7	3 × 10^−22^	intron	0.488	0.123	7.16 × 10^−5^	PVRL2	rs283813	1	7.6 × 10^−28^	33589840
8	TOMM40	rs112849259	19	44,894,050	C/t	2.6	0.73	missense	1.227	0.138	4.59 × 10^−19^					
9	TOMM40	rs741780	19	44,901,174	T/c	43.2	7 × 10^−2^	intron	−0.454	0.080	1.14 × 10^−8^	TOMM40	rs741780	1	1.4 × 10^−8^	23565137
10	APOE	rs440446	19	44,905,910	G/c	35.8	2 × 10^−2^	missense	−0.561	0.087	1.22 × 10^−10^	APOE	rs439401	0.57	1.1 × 10^−78^	21460841
11	APOE	rs429358	19	44,908,684	T/c	15.4	0.54	missense	1.371	0.077	4.63 × 10^−71^	APOE	rs429358	1	2.7 × 10^−78^	21390209
12	APOE	rs7412	19	44,908,822	C/t	8.0	0.66	missense	−0.856	0.200	1.87 × 10^−5^	APOE	rs7412	1	5.5 × 10^−58^	20885792
13	APOC4; APOC2	rs5167	19	44,945,208	T/g	35.1	0.10	missense	0.315	0.075	2.92 × 10^−5^	APOC4; APOC2	rs5167	1	2.8 × 10^−9^	21460840

^1^ Multiple genes were assigned if the index SNP was within the region of overlapping genes. ^2^ One single nucleotide polymorphism (**SNP**) per gene locus was retained. **Chr** = chromosome; **Ref/Alt** = reference/alternative allele (majuscule/minuscule letters stay for major/minor allele); **MAF obs** = minor allele frequency observer in our dataset. **Beta/SE/P** indicates effect size/standard error/*p*-value of the association of the alternative (minor) allele vs. the reference (major) allele. **SNP_GRASP_** and **Gene_GRASP_** denote SNP and related gene, for which minimum *p*-value has been reported in either the GRASP [32] or GWAS [33] catalog, where **P_GRASP_** is the reported *p*-value of the association with AD risk and **PMID** is the PubMed index of the respective paper. **r^2^** represents LD between previously reported SNP and SNP reported here. Empty cells: no index SNPs from the corresponding genetic locus were previously reported neither in the GRASP nor in GWAS catalog for their association with AD risk.

**Table 4 genes-14-01834-t004:** Univariate and pair-wise pleiotropic associations of SNP, which includes Alzheimer’s disease and one of 16 considered phenotypes: 4 genetic loci outside the APOE gene region.

N	Gene(s) ^1^	SNP ^2^	Chr	Location, Base Pairs GRCh38	Ref /Alt	MAF obs (%)	PH	Beta	SE	P	P_F_
1	CDK11B (MMP23B)	rs28688376	1	1,637,577	T/c	26.1	BMI	−0.096	0.017	3.36 × 10^−8^	1.42 × 10^−10^
2	CDK11B (MMP23B)	rs28688376	1	1,637,577	T/c	26.1	Weight	−0.232	0.052	7.01 × 10^−6^	2.37 × 10^−8^
3	OBP2B	rs11244035	9	133,205,932	C/t	10.2	LDL-C	1.206	0.187	1.09 × 10^−10^	1.13 × 10^−12^
4	OBP2B	rs11244035	9	133,205,932	C/t	10.2	TC	1.519	0.243	3.84 × 10^−10^	3.81 × 10^−12^
5	TPM1	rs4775613	15	63,056,897	A/g	42.8	TC	−0.436	0.146	2.81 × 10^−3^	9.91 × 10^−7^
6	TPM1	rs4775613	15	63,056,897	A/g	42.8	HDL-C	−0.245	0.048	2.43 × 10^−7^	1.31 × 10^−10^
7	TPM1	rs4775613	15	63,056,897	A/g	42.8	SBP	0.156	0.062	1.18 × 10^−2^	3.84 × 10^−6^
8	TPM1	rs4775613	15	63,056,897	A/g	42.8	HF	−0.052	0.026	4.16 × 10^−2^	1.24 × 10^−5^
9	SMARCA4	rs28997580	19	11,013,062	C/t	0.9	LDL-C	−5.450	0.591	2.81 × 10^−20^	<5 × 10^−8^
10	SMARCA4	rs28997580	19	11,013,062	C/t	0.9	TC	−6.141	0.767	1.20 × 10^−15^	<5 × 10^−8^

^1^ Multiple genes were assigned if the index SNP was within the region of overlapping genes. ^2^ One single nucleotide polymorphism (**SNP**) per gene locus was retained. **Chr** = chromosome; **Ref/Alt** = reference/alternative allele (majuscule/minuscule letters stay for major/minor allele); **MAF obs** = minor allele frequency **obs**erver in our dataset. **Beta/SE/P** indicates effect size/standard error/*p*-value of the association of the alternative (minor) allele vs. the reference (major) allele. **P_F_** is the *p*-value of the pair-wise pleiotropic analysis obtained by Fisher’s method.

## Data Availability

This manuscript was prepared using limited-access phenotypic and genetic datasets of the UK Biobank, which are available through the following link: http://www.ukbiobank.ac.uk/, accessed on 15 March 2021.

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
