# Peer review of "Exome-Wide Association Study Identified Clusters of Pleiotropic Genetic Associations with Alzheimer’s Disease and Thirteen Cardiovascular Traits"

_genes, 2023, doi:10.3390/genes14101834_

Round 1

Reviewer 1 Report

The study: Exome-Wide Association Study Identified Clusters of Pleiotropic Genetic Associations with Alzheimer’s Disease and Thirteen Cardiovascular Traits.is well-written and provides new insights into the disease. The results are clear and not misleading, the materials and methods section is clear and unambiguous. 

Introduction

"This means that there are other..." it would be interesting to mention the immune system because it is in contact withthe BBB and all other cell types 10.1124/pharmrev.121.000400 .

Additionally, 3 new treatments have recently received FDA approval or partial approval and it would  be interesting to mention them and discuss them in the discussion section in light of your findings. 

please clearly state the hypothesis along the goals of the study.

Material and methods:

Please provide a rational for analyzing only women as it will assist the reader and clarify the data. 

Maybe I missed it but I don't see which form of AD the patients have. This information is critical because the PSEN1 or APOE4 forms have different characteristics and the progression is not the same.

Additionally, please explain how the patients were selected from the database and detail the inclusion and exclusion criteria. 

Results:

The results are interesting and show that AD is not just in the brain. It's clearer than ever that AD is multifactorial. 

It would be interesting to include the inflammatory status of the patients. Inflammation (central or peripheral) has been linked to the acceleration or induction of AD. Given that inflammation is present in all the conditions studied in this article. That is why it would be valuable to provide data on the inflammatory status. 

Discussion:

Please discuss the limitation of this study, and the potential implication for the field. How can this study be applied in future research?

"This finding suggests that protective effects of higher BMI level against AD are related to higher weight rather than smaller height" I agree with this statement, but it raises some questions. High BMI is often associated with organs distress, high blood pressure, diabetes and inflammation. It would be interesting to discuss/explore  this point further. 

The manuscript is well-written and really interesting. The results are not misleading and are presented clearly. I will accept after minor revision. 

Thanks

Author Response

We would like to thank the editors and reviewers for their insightful comments and suggestions, which helped us to improve the manuscript. We revised the manuscript accordingly. Please find below our point-by-point answers to the reviewers’ comments including changes we made to address reviewers’ suggestions. Those modifications were highlighted in the revised version of the manuscript sent in a separate file.
In addition to changes made after addressing comments and suggestions provided by the reviewers, we corrected a misprint in Section 3.1. We replaced “187,818” with 188,260 that is the sample size used in this study. The previous number in that part, “187,818”, was a misprint.
Also, we would like to ask for landscape orientation for Table 3. In landscape orientation, this table will represent information in a more readable format.
Sincerely yours,
Authors
REVIEWER 1
The study: Exome-Wide Association Study Identified Clusters of Pleiotropic Genetic Associations with Alzheimer’s Disease and Thirteen Cardiovascular Traits.is well-written and provides new insights into the disease. The results are clear and not misleading, the materials and methods section is clear and unambiguous.
Introduction
[Comment 1] "This means that there are other..." it would be interesting to mention the immune system because it is in contact with the BBB and all other cell types 10.1124/pharmrev.121.000400 .
[Answer] We thank the reviewer for this reference and for drawing our attention to the immune system, which role was not mentioned in the Introduction section. We referred to the immune system response in the mentioned sentence and added corresponding reference, Ref.[5] in the revised version of the manuscript. [5] Pons V, Rivest S. Targeting Systemic Innate Immune Cells as a Therapeutic Avenue for Alzheimer Dis-ease. Pharmacol Rev. 2022;74:1-17.
[Comment 2] Additionally, 3 new treatments have recently received FDA approval or partial approval and it would be interesting to mention them and discuss them in the discussion section in light of your findings. [Answer] Thank you for this suggestion. We added a sentence and a new Ref. [3], J. Cummings, Drugs (2023), after the first sentence in the second paragraph of the Introduction section: [3] Cummings J. Anti-Amyloid Monoclonal Antibodies are Transformative Treatments that Redefine Alz-heimer's Disease Therapeutics. Drugs. 2023;83:569-76. “The amyloid beta hypothesis was extensively explored for clinical treatment of AD, which targets Aβ plaques. Two drugs (lecanemab, aducanumab) were approved by FDA through the accelerated approval mechanism and third one, donanemab, showed promising results [3], but there is no cure for AD till now.”
Additionally, we added a sentence in the Discussion section (at the end of section 4.1), which emphasized possible direction of developing new AD treatments considering our findings: “Our findings suggest additional mechanisms contributing to the disease initiation and/or progression as compared with the amyloid-beta treatment and emphasize actin filaments, chromatin remodeling, neuronal apoptosis and lipid abnormalities as possible targets for developing new AD drugs.”
[Comment 3] please clearly state the hypothesis along the goals of the study.
[Answer] We have already stated our hypothesis in the following sentence (see Introduction):
“We believe that this approach can more accurately characterize complex roles of genetic and non-genetic factors in AD pathogenesis [15-18], including complex interactions, which can be attributed to different biological mechanisms driving antagonistic genetic heterogeneity [19, 20].”
In order to clearly emphasize this, we introduced an additional sentence (see the last paragraph in the Introduction section): “We hypothesize that using summary statistics of pleiotropic associations of AD and related phenotypes (risk factors) can capture clusters of phenotypes according to similarity of genetic effects rather than according to direct correlation(s) between these phenotypes, which can help to identify shared biological pathways.”
Material and methods:
[Comment 4] Please provide a rational for analyzing only women as it will assist the reader and clarify the data.
[Answer] We apologize. We clarified that in this study a combined sample of men and women was investigated, see first paragraph in section 3.1. Additionally, we added similar clarification to the Methods section in the revised version of the manuscript. Now, the first sentence in section 2.2 reads: “Data from the UK Biobank (UKB) [26, 27] on individuals of Caucasian ancestry, men and women combined, were considered in the analyses.”
[Comment 5] Maybe I missed it but I don't see which form of AD the patients have. This information is critical because the PSEN1 or APOE4 forms have different characteristics and the progression is not the same.
[Answer] At the end of the first paragraph in the subsection “2.4 Phenotypes”, we indicated that disease status, including AD status, was defined by the UK biobank through the ICD-9 and ICD-10 codes:
“All cases were defined by the UK Biobank based on ICD-9 and ICD-10 codes.”
In order to clarify the AD status definition, we introduced an additional sentence at the end of this paragraph clearly stating which ICD-10 codes were used for definition of AD status: “AD status was defined by using codes F00 and G30 according to ICD-10 classification.”
These codes include early and late onset of AD.
[Comment 6] Additionally, please explain how the patients were selected from the database and detail the inclusion and exclusion criteria.
[Answer] We added explanation about definition of AD status (see previous answer). No exclusion criteria were applied. We did not apply any selection criteria related to age, sex, or other health conditions Because (1) AD cases included early (small proportion, 17 AD cases with age at onset younger than 60 years) and late onset of Alzheimer’s, (2) our study was focused on 16 risk factors and their genetic
components shared with AD, and (3) this study did not address any questions about causal link of AD and considered phenotypes, which would be applied otherwise.
Results:
[Comment 7] The results are interesting and show that AD is not just in the brain. It's clearer than ever that AD is multifactorial.
It would be interesting to include the inflammatory status of the patients. Inflammation (central or peripheral) has been linked to the acceleration or induction of AD. Given that inflammation is present in all the conditions studied in this article. That is why it would be valuable to provide data on the inflammatory status.
[Answer] We thank the reviewer for this suggestion. We agree with the reviewer that inflammation is one of several important factors related to AD [Migliore L, Coppede F. Gene-environment interactions in Alzheimer disease: the emerging role of epigenetics. Nat Rev Neurol. 2022;18(11):643-60.]. The effects of these factors require separate extensive consideration and analysis and will be published elsewhere.
Discussion:
[Comment 8] Please discuss the limitation of this study, and the potential implication for the field. How can this study be applied in future research?
[Answer] We added an implication of our findings related to the development of new AD treatments, see our answer to Comment 2.
Also, we added the following sentences at the end of the fourth paragraph in the Conclusion section: “Defining more homogeneous patterns of phenotypes and genetic factors can help to identify more homogeneous groups of people who are at different risks for AD. Further research can help to identify genetic and non-genetic components in pleiotropic effects related to Alzheimer’s disease, which can help combine and differentiate treatments involving drug therapy and life-style changes.”
Additionally, we listed limitations of our study at the end of the Conclusion section: “This study has some limitations. (1) Analyses were performed for the entire sample but not separately in each sex group and/or distinct age groups, i.e. for individuals younger or older than 65 years. (2) We used a threshold p-value = 5E-4 for associations with AD, which substantially limited the number of significant pleiotropic associations/SNPs. (4) Small number of SNPs (because of limitation (3)) were available for hierarchical cluster analysis, which limited precision of identifying phenotype clustering.”
[Comment 9] "This finding suggests that protective effects of higher BMI level against AD are related to higher weight rather than smaller height" I agree with this statement, but it raises some questions. High BMI is often associated with organs distress, high blood pressure, diabetes and inflammation. It would be interesting to discuss/explore this point further.
[Answer] Thank you for your suggestion about exploring links between BMI/weight and AD risk. In order to address this point, we included the following sentence at the end of subsection 4.2: “While increased BMI, i.e. increased weight, is associated with higher risk of cardio-vascular diseases (CHD, MI, hypertension) and diabetes in middle life, decreased BMI and weight in late life can increase AD risk because of deteriorated functions of an aging organism, which includes deteriorated metabolism, decreased cholesterol level, decreased muscle mass, frailty, sarcopenia etc.”
[Comment 10] The manuscript is well-written and really interesting. The results are not misleading and are presented clearly.
[Answer] Thank you for your overall positive opinion about our study and suggestions that we believe helped us to improve the manuscript considerably.

Reviewer 2 Report

This study employs the UKB cohort to explore genetic variants associated with Alzheimer's Disease (AD), as well as shared genetic variants between AD and 16 cardiovascular phenotypes. Given the established relationship between cardiovascular risk factors and AD, this approach provides an intriguing avenue to elucidate AD pathology. However, several questions arise concerning the study design and statistical analysis. My comments are outlined below:

Major Comments:

1. Introduction

I recommend that the authors offer a comprehensive review of the comorbidity between AD and the selected 16 cardiovascular risk factors. This would provide clearer context for the study. While Page 2, Paragraph 2 mentions "Pleiotropic GWAS and EWAS" in relation to LDL-C, TC, and HDL-C as risk factors, this term lacks precise definition and is not widely used. The authors should provide a clear definition if they opt to use this terminology. Additionally, the cited references (4-6) appear to be observational researches rather than genetic investigations. So those examples don’t actually provide any insights to the potential of shared genetics between AD and those phenotypes.

2. Methods

Section 2.2 and Table 1: It would enhance the paper to include demographic information for the AD group and the non-AD (control) group, separately. It would also be informative to report any statistically significant differences in age or sex between these groups.

Section 2.3: The manuscript should elaborate on the quality control processes, variant calling, and variant selection, as these steps form the foundation for the study and any subsequent replication efforts.

Section 2.5: A correction for multiple comparisons should be applied.

Section 2.6: Please furnish an introductory explanation and appropriate citations for the Fisher’s method in this context.

Section 2.8: The paper notes different p-value thresholds for AD and cardiovascular traits in the EWAS univariate associations. Clarification regarding the rationale behind these different thresholds would be beneficial.

Section 2.9: The criteria for defining the index SNP, based on its Minor Allele Frequency (MAF) being closer to MAF from UK10K TWINS and/or the 1000 Genomes reference datasets, seems insufficient. Given that the purpose of an index SNP is to identify likely causal variants while avoiding false positives influenced by linkage disequilibrium, this approach may warrant reconsideration.

Section 2.10: Would the authors be able to furnish additional evidence to substantiate the validity of the methodology employed?

3. Results

Genetic Correlation: Given that summary statistics from EWAS are available, could the authors calculate the genetic correlation for each pair of AD and cardiovascular phenotypes? Understanding the genetic correlation could offer valuable insights into their relationship.

APOE Gene Cluster Associations (Section 3.2): The paper mentions 20 associations within the APOE gene cluster that were not driven by genome-wide significance in their relationship with AD. Was any specific test conducted to substantiate this assertion? If this conclusion is solely based on a lack of univariate association with AD, it may be worth reconsidering its validity, especially given how index SNPs are defined.

Index SNPs in Cluster Analysis: My understanding is that only index SNPs were used in the cluster analysis. Could the authors specify how many candidate SNPs within a locus exhibiting antagonistic genetic heterogeneity also showed the same pattern?

4. Discussion

Interpreting Results: A discussion that synthesizes the current findings with existing knowledge of comorbidities between AD and cardiovascular phenotypes would provide a comprehensive understanding of the study’s implications.

Minor Comments:

1. Incorporating a flowchart could help delineate the study design more clearly and aid in reader comprehension.

2. I recommend avoiding unnecessary abbreviations. For instance, “GW” in the main text and “H” and “W” in the figures and tables could be spelled out for clarity.

3. It appears that Section 2.7 is absent from the manuscript.

4. Annotations explaining the color legend in the figures should be included for clarity.

See my comments above.

Author Response

We would like to thank the editors and reviewers for their insightful comments and suggestions, which helped us to improve the manuscript. We revised the manuscript accordingly. Please find below our point-by-point answers to the reviewers’ comments including changes we made to address reviewers’ suggestions. Those modifications were highlighted in the revised version of the manuscript sent in a separate file.

In addition to changes made after addressing comments and suggestions provided by the reviewers, we

corrected a misprint in Section 3.1. We replaced “187,818” with 188,260 that is the sample size used in this study. The previous number in this part, “187,818”, was a misprint.

Also, we would like to ask for landscape orientation for Table 3. In landscape orientation, this table will represent information in a more readable format.

Sincerely yours,

Authors

REVIEWER 2

This study employs the UKB cohort to explore genetic variants associated with Alzheimer's Disease (AD), as well as shared genetic variants between AD and 16 cardiovascular phenotypes. Given the established relationship between cardiovascular risk factors and AD, this approach provides an intriguing avenue to elucidate AD pathology. However, several questions arise concerning the study design and statistical analysis. My comments are outlined below:

Major Comments:

  1. Introduction

[Comment 11] I recommend that the authors offer a comprehensive review of the comorbidity between AD and the selected 16 cardiovascular risk factors. This would provide clearer context for the study.

[Answer] We thank the reviewer for this constructive suggestion. Review on comorbidities with AD is a separate extensive topic and it is out of the scope of this paper.

[Comment 12] While Page 2, Paragraph 2 mentions "Pleiotropic GWAS and EWAS" in relation to LDL-C, TC, and HDL-C as risk factors, this term lacks precise definition and is not widely used. The authors should provide a clear definition if they opt to use this terminology.

[Answer] We introduced definition for pleiotropic EWAS and GWAS after the sentence where univariate GWAS and EWAS were mentioned. This additional sentence reads as follows:

“While conventional GWAS and EWAS focus on identifying genetic associations with single traits, pleiotropic GWAS and EWAS take a broader approach. They aim to simultaneously assess genetic associations with multiple traits and identify shared genetic components that contribute to the pleiotropy observed in epidemiological studies.”

[Comment 13] Additionally, the cited references (4-6) appear to be observational researches rather than genetic investigations. So those examples don’t actually provide any insights to the potential of shared genetics between AD and those phenotypes.

[Answer] We thank the reviewer for this comment. We have added appropriate references, Refs[9,10] in the revised version, and rewritten related sentences in order to clearly express our point. In the revised versions of the manuscript, the entire paragraph reads as follows:

“Genome- (GWAS) and exome- (EWAS) wide association studies are comprehensive tools designed to identify genes and related biological mechanisms associated with different traits. While conventional GWAS and EWAS focus on identifying genetic associations with single traits, pleiotropic GWAS and EWAS take a broader approach. They aim to simultaneously assess genetic associations with multiple traits and identify shared genetic components that contribute to the pleiotropy observed in epidemiological studies. Previous epidemiological studies, which considered AD and other related traits, helped identify several cardiovascular factors conferring AD risk. For instance, increased levels of LDL-C and TC are considered as risk factors of AD [6, 7], while increased level of HDL-C was highlighted as a protective one [8]. Pleiotropic GWAS/EWAS provided an additional opportunity to identify several genetic components and respective biological mechanisms shared by those multiple traits [9, 10].”

[9] Bone WP, Siewert KM, Jha A, Klarin D, Damrauer SM, Program VAMV, et al. Multi-trait association studies discover pleiotropic loci between Alzheimer's disease and cardiometabolic traits. Alzheimers Res Ther. 2021;13:34.

[10] Broce IJ, Tan CH, Fan CC, Jansen I, Savage JE, Witoelar A, et al. Dissecting the genetic relationship between cardiovascular risk factors and Alzheimer's disease. Acta Neuropathol. 2019;137:209-26.

  1. Methods

[Comment 14] Section 2.2 and Table 1: It would enhance the paper to include demographic information for the AD group and the non-AD (control) group, separately. It would also be informative to report any statistically significant differences in age or sex between these groups.

[Answer] We thank the reviewer for this suggestion. We updated Table 1 by presenting demographic information for groups of AD cases and controls separately, and highlighted significance of respective differences between the groups at the end of Section 2.4:

“We found significant differences between AD cases and controls in both qualitative phenotypes (assessed using the exact Fisher test) and mean values for quantitative phenotypes (evaluated using the Wald test) across all traits.”

The following sentence was added to the notes in Table 1:

p-values for qualitative and quantitative phenotypes were assessed by using the exact Fisher and Wald test, respectively.“

[Comment 15] Section 2.3: The manuscript should elaborate on the quality control processes, variant calling, and variant selection, as these steps form the foundation for the study and any subsequent replication efforts.

[Answer] We extended subsection “2.3. Genotypes” with a description of quality of selected SNPs “213,935 SNPs were with a missing call rate better than 5%. For these SNPs, all individuals had a missing call rate better than 5 %. Only these SNPs were considered for reporting results of this study. We did not apply Hardy-Weinberg equilibrium test at this stage because negligible deviation from Hardy-Weinberg equilibrium was still highly significant in the large UKB sample.”

[Comment 16] Section 2.5: A correction for multiple comparisons should be applied.

[Answer] We did not apply any correction for multiple testing (adjustment of p-values) when considering correlations between phenotypes because we do not perform any hypothesis-free testing and we were interested in correlations only.

[Comment 17] Section 2.6: Please furnish an introductory explanation and appropriate citations for the Fisher’s method in this context.

[Answer] We extended explanatory text that describes Fisher’s method explaining multiple testing and provided proper reference. In the revised version of the manuscript, the modified text reads as follows:

“Given small correlation between traits (Table 2), Fisher’s method [31] was used for pleiotropic analysis. It combines p-values across phenotypes disregarding the effect directions and correlation between them and addresses the issue of multiple testing by increasing the number of degrees of freedom.”

[31] Fisher RAS. Statistical methods for research workers. 14th ed. ed. Edinburgh: Oliver and Boyd; 1970.

[Comment 18] Section 2.8: The paper notes different p-value thresholds for AD and cardiovascular traits in the EWAS univariate associations. Clarification regarding the rationale behind these different thresholds would be beneficial.

[Answer] Thank you for this suggestion. We extended respective sentences where selection of p-values for associations with AD and each of other phenotypes. The following sentence was added at the end of subsection 2.8:

“As this analysis was AD-centric we applied more stringent threshold for AD associations and less stringent threshold for each of the other phenotypes.”

[Comment 19] Section 2.9: The criteria for defining the index SNP, based on its Minor Allele Frequency (MAF) being closer to MAF from UK10K TWINS and/or the 1000 Genomes reference datasets, seems insufficient. Given that the purpose of an index SNP is to identify likely causal variants while avoiding false positives influenced by linkage disequilibrium, this approach may warrant reconsideration.

[Answer] We thank the reviewer for this comment. We clarified the selection of index SNPs and our previous reference to UK10K TWINS and/or the 1000 Genomes reference datasets. The revised version of this sentence reads as follows:

“The index SNPs were selected based on the most significant association (smallest p-values) within each genetic locus. Furthermore, the selection of index SNPs took into consideration the proximity of MAF from the UKB exome chip to MAF found in the UK10K TWINS and/or 1000 G reference datasets, as MAF for some SNPs substantially differed.”

[Comment 20] Section 2.10: Would the authors be able to furnish additional evidence to substantiate the validity of the methodology employed?

[Answer] In order to substantiate the validity of our approach we added the following text at the beginning of Section 2.10:

“Hierarchical cluster analysis can be applied if a distance measure is defined. Previous studies [11-17] used individual-level data of phenotype measurements. Such distance measure, which assesses similarity of phenotypes, includes genetic effects, effects of exogenous exposures and their interaction. In this study we propose to use genetic associations with phenotypes as a distance measure. This approach is more appropriate to gain insight into similarities related to genetic components rather than exogenous exposures considering that the contributions of genetic effects and exogenous exposures on correlations between the phenotypes can be substantially different (see Table 2). In this study, …”

Additionally, we would like to emphasize that summary statistics, i.e. effect sizes, define Euclidian (multi-dimensional) space. Hierarchical cluster analysis is well defined on Euclidian (multi-dimensional) space and previously was widely used in the field (see Refs.[11-17] in the revised version of the manuscript), which additionally justify our approach.

  1. Results

[Comment 21] Genetic Correlation: Given that summary statistics from EWAS are available, could the authors calculate the genetic correlation for each pair of AD and cardiovascular phenotypes? Understanding the genetic correlation could offer valuable insights into their relationship.

[Answer] We calculated the correlation based on 213,935 QC SNPs from the summary statistics of univariate EWAS analyses. In the revised manuscript, the results of correlations between summary statistics of genetic associations with AD and each of the other traits are presented in Table 2.

We added one subsection, subsection 2.6 in the revised version of the manuscript, which describes correlations between summary statistics to highlight the update.

“2.6 Correlation between summary statistics

Table 2 also includes information on Pearson correlation coefficients that were obtained by using summary statistics of genetic associations with AD and each of the considered quantitative and qualitative phenotypes. Only correlation coefficient between summary statistics for AD and LDL-C was statistically non-significant. Correlation coefficients of summary statistics for other traits with AD were statistically significant. Moreover, for five traits (MI, stroke, HF, weight and SBP), the correlation based on individual-level data and on summary statistics demonstrated opposite directions and were statistically significant. This demonstrates that correlations between phenotypes at individual-level can be shaped by other factors in addition to genetic component(s).”

Title of Table 2 was updated accordingly. Now it reads:

“Table 2. Pearson correlation coefficient between Alzheimer's disease and each of the considered phenotypes in the UK Biobank based on individual-level data and summary statistics of genetic associations.”

[Comment 22] APOE Gene Cluster Associations (Section 3.2): The paper mentions 20 associations within the APOE gene cluster that were not driven by genome-wide significance in their relationship with AD. Was any specific test conducted to substantiate this assertion? If this conclusion is solely based on a lack of univariate association with AD, it may be worth reconsidering its validity, especially given how index SNPs are defined.

[Answer]

In the referred sentence (i.e. “They included 20 associations in the APOE gene cluster, which were not driven by GW significance of the associations with AD.”), these 20 SNPs demonstrated genome-wide significant association with AD, i.e. with p-values < 5 x 10^-8, and significant associations with at least one of the other phenotypes. We excluded from the results reporting SNPs when they demonstrated GW significant association with AD and at least one GW significant pleiotropic associations, but their association with each of the other phenotypes, i.e. excluding AD< were nonsignificant, i.e. p-values >0.5.

NOTE: In pair-wise analysis, Fisher’s test provided lower p-values for pleiotropic associations for two phenotypes if each of the two associations is significant at the level of p-value <0.05. This observation was used for selecting associations when SNP was associated with AD at p-value < 5 x 10^-8, because for such SNPs pleiotropic test will provide smaller p-value if the second association was significant at the level of p-value < 0.05.

[Comment 23] Index SNPs in Cluster Analysis: My understanding is that only index SNPs were used in the cluster analysis. Could the authors specify how many candidate SNPs within a locus exhibiting antagonistic genetic heterogeneity also showed the same pattern?

[Answer] This is correct. Only index SNPs were used in the cluster analysis. If non-index SNP is in LD with index SNP it shows the same pattern of antagonistic associations. There was only one pattern of associations in the locus and this pattern was captured by the index SNP.

  1. Discussion

[Comment 24] Interpreting Results: A discussion that synthesizes the current findings with existing knowledge of comorbidities between AD and cardiovascular phenotypes would provide a comprehensive understanding of the study’s implications.

[Answer] We thank the review for this suggestion. We would like to emphasize that a few last paragraphs at the end of Section 4.2 (See Discussion) have already discussed this question:

“Four traits (HT, SBP, DBP, DM) defined a cluster with complex pattern of associations. As they are risk factors for CHD and MI, they are typically considered together with these diseases. Meanwhile, in this study, these phenotypes belong to two different clusters as there is more similarity within clusters than between clusters. All these six phenotypes and height would belong to the same cluster if the cut-off of the cluster height level is increased to 7 (Figure 21, see the vertical axis with numbers in the top left corner). Interestingly, SNPs encoding APOE ε2 and ε4 alleles were associated with two different subclusters, (HT, SBP) and (DBP, DM), respectively. This finding emphasizes that pleiotropic as-sociations of these APOE alleles have different etiology.

Height and (BMI, weight) defined two non-overlapping clusters indicating a role of independent mechanisms contributing to height and BMI or weight and AD.”

In order to substantiate further, we added several sentences in the main text as listed below.

We explored links between BMI/weight and AD risk and included the following sentence at the end of subsection 4.2:

“While an increased BMI resulting from weight gain is associated with a higher risk of cardiovascular diseases (such as CHD, MI, and hypertension) and diabetes in midlife, a decreased BMI resulting from weight loss in later life can increase the risk of AD. This risk is attributed to the deteriorated functions of an aging organism, which include impaired metabolism, reduced cholesterol levels, decreased muscle mass, frailty, and sarcopenia, among other factors.”

We added a sentence in the Discussion section (at the end of section 4.1), which emphasized possible direction of developing new AD treatments considering our findings:

“Our findings suggest the involvement of other mechanisms contributing to the disease pathogenesis in addition to amyloid-beta mechanism. Specifically, they highlight actin filaments, chromatin remodeling, neuronal apoptosis, and lipid abnormalities as potential targets for the development of new AD drugs.”

Also, we added the following sentences at the end of the fourth paragraph in the Conclusion section:

“Defining more homogeneous patterns of phenotypes and genetic factors can aid in the identification of more homogenous groups of individuals who are at varying risks for AD. Subsequent research can facilitate the identification of genetic and non-genetic components contributing to pleiotropic effects associated with AD. Such insights can be valuable for developing comprehensive treatment approaches that combine drug therapy and life-style interventions.”

Minor Comments:

[Comment 25]

  1. Incorporating a flowchart could help delineate the study design more clearly and aid in reader comprehension.

[Answer] We provided a flowchart for our study. It was added into the Main Text as Figure 1 in the revised version of the manuscript. Two other Figures were renumbered accordingly. Additionally, Figure caption and respective description for this new figure were added into the main text,

The following Caption was added to this new figure:

“Figure 1. Flowchart of the analyses of this Exome-Wide Association Study (EWAS) study.”

The following sentence was added to the main text at the beginning of the Results section. referring to this new figure:

“Figure 1 represents flowchart of the analyses performed in this study.”

[Comment 26]

  1. I recommend avoiding unnecessary abbreviations. For instance, “GW” in the main text and “H” and “W” in the figures and tables could be spelled out for clarity.

[Answer] We thank the reviewer for this suggestion. We made respective corrections. In the revised version of the manuscript, we spelled out weight and height in Table 4, Figures 2 and 3. We think that using “GW” abbreviation in the “Results” section is appropriate because this abbreviation was used more than 10 times and widely used in papers published by other researchers. In the revised version of the manuscript, we spelled out “GW” abbreviation in the Discussion section where, we think, it is more appropriate.

[Comment 27]

  1. It appears that Section 2.7 is absent from the manuscript.

[Answer] We thank the reviewer for this comment. We added one subsection, 2.6, and renumbered subsequent subsections in Section 2.

[Comment 28]

  1. Annotations explaining the color legend in the figures should be included for clarity.

[Answer] We think that description of color’s meaning in the figure captions is more appropriate than in the figures themselves and will complicate but not provide more clarification.